# The Impact of Regular Physical Exercise on Psychopathology, Cognition, and Quality of Life in Patients Diagnosed with Schizophrenia: A Scoping Review

**DOI:** 10.3390/bs13120959

**Published:** 2023-11-21

**Authors:** Lucía Vila-Barrios, Eduardo Carballeira, Adrián Varela-Sanz, Eliseo Iglesias-Soler, Xurxo Dopico-Calvo

**Affiliations:** Performance and Health Group, Department of Physical Education and Sport, University of A Coruna, 15179 A Coruña, Spain; lucia.vila.barrios@udc.es (L.V.-B.); adrian.varela.sanz@udc.es (A.V.-S.); eliseo.iglesias.soler@udc.es (E.I.-S.); xurxo.dopico@udc.es (X.D.-C.)

**Keywords:** schizophrenia, psychopathology, quality of life, cognition, physical exercise

## Abstract

The presence of less healthy lifestyle habits among individuals diagnosed with schizophrenia which can contribute to the escalation of physical disorders and exacerbation of psychological symptoms is well documented. The present scoping review aims to synthesize and evaluate the available evidence regarding the impact of regular physical exercise on psychopathology, cognition, and quality of life (QoL) in patients diagnosed with schizophrenia. A literature search was performed across Web of Science, SCOPUS, PubMed, and SPORTDiscus for randomized control trials published up to April 2022. Two independent reviewers applied the selection criteria and a third reviewer resolved discrepancies. A total of twelve studies were included, of which nine used endurance training and three used concurrent training (one of these additionally used resistance training). The results reveal benefits of various modalities of supervised regular exercise in the psychopathology of schizophrenia. Furthermore, regular endurance training seems to improve cognitive function in patients with schizophrenia and promote their QoL; however, results are inconclusive with respect to this last variable. The assessment of methodological quality in the reviewed articles indicates a high overall risk of bias, particularly in relation to deviations from intended interventions and the selection of reported results. Furthermore, an assessment of exercise reporting revealed that only 5 out of 19 items were fulfilled in more than 50% of the articles. Future research is needed to evaluate the effects of different training modalities and the optimal dose–response relationship in patients diagnosed with schizophrenia.

## 1. Introduction

According to the International Classification of Diseases (ICD-11), schizophrenia is a mental disease that produces alterations in different brain areas affecting mental health, including thinking, self-experience, perception, volition, and occasionally perturbing behavior [1]. In addition, these alterations commonly concur with associated symptoms such as delusions, hallucinations, disorganized speech or behavior, and decreased emotional expression [2]. Updated prevalence data show that schizophrenia affected 23.59 million people (0.32%) worldwide in 2019, while in Europe this prevalence rises to 2.8 million people (0.35%) [3]. Further, this mental illness is associated with a high economic burden regarding public health [4,5]. For instance, the economic burden of schizophrenia in the United States is estimated to be nearly USD 150 billion [5,6] entailing more than USD 12,000 per patient when hospitalization is required [7]. In England, that annual cost in 2012 was approximately GBP 11.8 billion, equivalent to around USD 15.8 billion [8]. 

One of the crucial factors that affect health-related domains other than the mental health (i.e., physical fitness or social behaviors) of individuals diagnosed with schizophrenia is their lifestyle patterns. Compared to the general population, people with schizophrenia tend to have less healthy habits. Specifically, they engage in less physical exercise, have a smoking prevalence that is about 50% higher, and follow a less healthy diet characterized by increased fat intake and decreased consumption of fruits and vegetables [9,10,11]. Scientific evidence has widely demonstrated predominantly sedentary behavior in the daily routine of individuals diagnosed with schizophrenia [12,13,14,15,16]. Specifically, between 50 and 70% of patients fail to meet the physical activity recommendations of at least 150 min of moderate intensity exercise per week, which classifies them as physically inactive [12,16]. This context, along with disease-specific circumstances such as the lack of motivation, cognitive and psychopathological symptoms, and medication side effects, make it difficult to reach an optimal level of physical activity [10,12,14,16]. Even though lower levels of physical activity (i.e., ≥90 min per week) may produce positive effects on overall cognition [17], it has been consistently demonstrated that physical inactivity and sedentary behavior lead to an increased risk of developing cardiovascular diseases and metabolic disorders in these patients [10,14,15,16], and can even aggravate some psychological symptoms.

In this sense, antipsychotics can aggravate this situation by causing weight gain [11,18]. This increase in body weight creates a detrimental cycle of effects, perpetuating the negative health outcomes [11,18] associated with antipsychotic use. This context leads to a reduction in life expectancy by approximately 10–20 years [11,16,19,20], as well as a two- to threefold rise in mortality prevalence compared to the general population [9,19,21]. The aforementioned situation underscores the need to contemplate the effect of this specific pathology on patients’ quality of life (QoL). The World Health Organization (WHO) defines QoL as an individual’s perception or their position in life in the context of the culture and value systems in which they live and concerning their goals, expectations, standards, and concerns [22]. Therefore, it is imperative to analyze how this pathology influences various aspects of an individual’s QoL in order to improve overall patient care and well-being. Previous studies have demonstrated that the deterioration of different health areas experienced by patients with schizophrenia can lead to a decrease in their QoL [23,24]. This situation can be aggravated with age, as the prevalence of chronic physical diseases such as hypertension or diabetes increases [23].

Considering the impact of schizophrenia on public health and individuals’ health and QoL, the treatment of this pathology is of particular interest. The commonly used pharmacological treatment of schizophrenia can be categorized into first-generation antipsychotics (e.g., haloperidol, levomepromazine, or chlorpromazine) and second-generation antipsychotics (e.g., clozapine, olanzapine, or risperidone) [25,26,27]. The objective of these drugs is to reduce symptoms and improve patients functioning. However, such treatments can cause various side effects, including extrapyramidal effects (i.e., dyskinesia, dystonia, akathisia, tremor, or rigidity), cardiovascular effects (i.e., tachycardia, palpitations or arrhythmias), and severe adverse effects that affect metabolism (i.e., obesity and diabetes) [25,26,27], as well as sexual dysfunction that can reduce patients’ QoL and satisfaction [27]. 

In light of these challenges, physical activity and regular exercise have emerged as a promising and effective alternative for mitigating collateral effects and improving psychopathological and cognitive symptoms, thereby promoting societal functioning and QoL [16,18]. As with pharmacological treatment, exercise should be carefully prescribed for its clinical application as well as to ensure its replicability, allowing researchers to better assess and compare the results and conclusions derived from scientific evidence [28]. In accordance with this, the American College of Sports Medicine (ACSM), in its 10th edition of Guidelines for Exercise Testing and Prescription (GETP10), described the six components that should be detailed and reported in the prescription of physical exercise: Frequency, Intensity, Time, Type, Volume of exercise and Progression component (FITT−VP) [29].

In recent years, a growing body of scientific research has focused on investigating the effects of physical exercise among people with schizophrenia. One recent study concluded that endurance exercise produces important health benefits with minimal negative effects [30]. Similarly, a review of combined endurance and resistance exercise interventions (i.e., concurrent training) for individuals with schizophrenia found that they promoted mental and physical health [31]. In another systematic review, the authors reported that regular exercise enhances physical fitness and lowers cardiometabolic risk factors [32]. Along the same lines, Bredin and colleagues [33] demonstrated the benefits of exercise training on psychiatric symptoms and health-related physical measures in an interesting meta-analysis. However, there is controversy about which exercise modalities have positive effects on QoL, cognition, and psychopathological symptoms [34,35]. 

While the scientific evidence suggests a favorable impact of exercise on health-related parameters in individuals diagnosed with schizophrenia, serious concerns arise regarding the comprehensive understanding of the most influential factors and the methods employed to ensure reproducibility. This is especially relevant when contemplating the description of the independent variable, that is, physical exercise. The reviewed literature lacks both clear inclusion criteria and adequate information pertaining to adherence to the FITT-VP parameters, which are essential for the purposes of prescribing exercise and facilitating the replicability of research findings [28,29]. These deficiencies persist in recent reviews published within the last year [36,37,38,39,40]. Thus, for this new review it is imperative to incorporate elements addressing these issues into the selection criteria.

On the other hand, certain reviews [31,32,33] have failed to differentiate between distinct pathologies, encompassing patients with an initial episode of psychosis. This lack of differentiation may introduce confounding factors when analyzing outcomes, especially considering that schizophrenia embodies a more intricate diagnostic construct characterized by severe deterioration and enduring symptoms [1]. Furthermore, it is essential to consider the severity of symptoms and the treatment settings, as emphasized by Cella et al. [36]. In fact, one of the reviews published in 2023 [37] concluded that the positive effects were more pronounced among outpatients. Taking these factors into consideration along with the observation that recent reviews in this area have failed to address this differentiation [36,37,38,39], it becomes imperative to direct our focus exclusively toward outpatients. Such an approach holds the potential to generate a more consistent study cohort, thereby enhancing the applicability of our findings to clinical practice. Therefore, this scoping review aims to synthesize and assess the available evidence regarding the effects of regular physical exercise on psychopathology, cognition, and QoL in outpatients diagnosed with schizophrenia.

## 2. Methods

### 2.1. Literature Search

The present scoping review protocol was registered on the Open-Source Framework (OSF) on 8 February 2023 (https://osf.io/tvbr6/ (accessed on 8 February 2023)) and followed the recommendations proposed by Tricco and colleagues using the Preferred Items for Systematic Reviews and Meta-Analyses extension for Scoping Reviews (PRISMA-ScR) [41].

The literature search was performed independently by three blinded investigators on 5 April 2022 across four electronic databases (Web of Science, SCOPUS, PubMed, and SPORTDiscus). The search strategy (Appendix A) used was a combination of keywords and Boolean operators, resulting in the following search equation: (“power exercise*” OR “strength exercise*” OR “resistance train*” OR “strength train*” OR “power train*” OR “resistance exercise*” OR “regular exercise” OR “exercise program” OR “physical activity” OR “endurance” OR “human walking” OR “physical exercise*” OR “fitness train*” OR “plyometric*” OR “aerobic exercise*” OR “aerobic train*” OR “muscle stretching” OR “tai chi*” OR “yoga” OR “physical condition*”) AND (“schizo*”). 

### 2.2. Eligibility Criteria

We followed a PICOS approach to select studies eligible for inclusion (Table 1). Studies in a language other than Spanish, English, or Portuguese were excluded. 

### 2.3. Study Selection and Data Extraction

Two researchers (L.V.B and A.V.-S.) independently and blindly screened records obtained from the electronic databases for data inclusion and identified and removed duplicated studies using Zotero [42]. Subsequently, an exhaustive manual review of the remainder articles was performed to avoid potential errors. After that, the same reviewers screened the articles with the help of Rayyan [43] by analyzing the title and abstract, identifying those that met the inclusion criteria, and removing those with characteristics described in the exclusion criteria. In the final stage, the same researchers independently conducted a comprehensive examination of the full text to determine the final eligibility of the studies. These latter two phases were overseen by a third reviewer (X.D.-C.), who resolved discrepancies that arose from disagreements due to the individual decisions made by the researchers. 

The following data were extracted from the included full-text publications (Table 2): (1) author and year of publication; (2) participants’ characteristics (sex, body mass index, race/ethnicity, pharmacological treatment, previous physical exercise experience); (3) design of the study, sample size, and intervention period; (4) exercise protocol, supervision, and training familiarization; and (5) outcomes of interest.

### 2.4. Critical Appraisal

To critically assess the studies included in the present review, we individually evaluated the methodological quality and quality of reporting of training protocols for each intervention.

#### 2.4.1. Risk of Bias

The risk of bias in the included studies was assessed according to the Revised Cochrane risk-of-bias tool for randomized trials (RoB 2) [44]. The following domains were scored for each included study: (1) bias arising from the randomization process; (2) bias due to deviations from intended interventions; (3) bias due to missing outcome data; (4) bias in the measurement of the outcome; and (5) bias in the selection of the reported results. Each of these domains was judged as having a *low risk* of bias, *some concerns*, and a *high risk* of bias, respectively [44].

#### 2.4.2. Quality Reporting of Exercise Intervention in the Training Programs

To assess the quality of the independent variable of the eligible studies, i.e., exercise intervention, we used the Consensus on Exercise Reporting Template (CERT) tool [28].

**Table 2 behavsci-13-00959-t002:** Characteristics of the included studies.

Study	Participants Characteristics	Design	Exercise Protocol	Outcomes of Interest
Andrade e Silva et al. [45]	***Sex***: 100% male ***BMI* (kg/m^2^):** 27.98 ± 1.67 (RESEX), 29.42 ± 1.93 (CONCEX), 25.38 ± 1.60 (CTRL), ***Race/Ethnicity:*** N/A ***Other diseases:*** N/A ***Pharmacological treatment:*** Yes, antipsychotics, antidepressants (n = 15), benzodiazepine (n = 6), mood stabilizer (n = 6) and anticholinergic (n = 8) *Previous physical exercise experience:* No (sedentary lifestyle > 1 year) Males with a DSM-IV diagnosis of schizophrenia, 18–50 years (33.36 ± 7.6 years) with stable doses of medication and clinically stable disease.	***Design***: RCT RESEX CONCEX CTRL ***Sample size:*** (n = 47) RESEX: n = 12/CONCEX: n = 9/CTRL: n = 13 ***Intervention period:*** 20 weeks	*Supervised:* Yes *Familiarization:* 3 sessions ***RESEX:*** *F:* 2 sessions/week *IL*: 40% to 85% 1RM *Ti*: 55 min/session *Ty*: progressive RT. *V*: 7 exercises from 2 × 15 reps to 3 × 6–8 reps (r: 1–2 min between sets). From 210 reps at week 1 to 147 reps at week 20 *P*: 2.5% per week ***CONCEX***: *F:* 2 sessions/week *IL*: RT: 40–85% 1RM/ET: 40–75% VO_2max_ *Ti*: 55 min/session *Ty*: progressive RT + progressive ET. *V*: RT: 7 exercises from 1 × 15 reps to 2 × 6–8 reps (r: 1–2 min), 98–105 reps per week/ET: 50 min per week *P*: RT: +2.5% per week (RT)/ET: +1.75% per week ***CTRL***: *F:* 2 sessions/week *IL*: minimum load *IE*: 7 exercise 2 × 15 reps (r: 1 min between sets) *Ti*: 55 min/session *Ty*: RT with minimum load *V*: 210 reps/week *P*: N/A *Attrition of the study*: 27.65% *Adherence to the study:* >75%	*Analysis within group* ***RESEX:*** ↓ PANSS total score after 10 weeks (*p* = 0.002) ↓ PANSS total score after 20 weeks (*p* ≤ 0.001) ↓ positive symptoms (PANSS) after 10 weeks (*p* = 0.039) ↓ positive symptoms (PANSS) after 20 weeks (*p* ≤ 0.001) ↓ negative symptoms (PANSS) after 10 weeks (*p* = 0.001) ↓ negative symptoms (PANSS) after 20 weeks (*p* = 0.002) ↑ SF- 36 (physical role dimension) after 20 weeks (*p* = 0.011) ***CONCEX:*** ↓ PANSS total score after 10 weeks (*p* = 0.026) ↓ PANSS total score after 20 weeks (*p* = 0.003) ↓ positive symptoms (PANSS) after 20 weeks (*p* = 0.016) = negative symptoms (PANSS) (*p* not reported) ↑ SF- 36 (physical role dimension) after 20 weeks (*p* = 0.014) ***CTRL:*** = PANSS total score (*p* not reported) = positive symptoms (PANSS) (*p* not reported) = negative symptoms (PANSS) (*p* not reported) *Analysis between groups* Not reported for these outcomes
Lo et al. [46]	***Sex:*** 17.64% male (HIIT)/50% male (ENEX)/50% male (CTRL) ***BMI* (kg/m^2^):** 26.72 ± 5.31 (HIIT)/25.56 ± 4.09 (ENEX)/24.82 ± 2.83 (CTRL) ***Race/Ethnicity:*** N/A ***Other diseases:*** N/A ***Pharmacological treatment:*** 16 were prescribed atypical antipsychotics and 1 typical antipsychotic in HIIT group/13 were prescribed atypical antipsychotics, 2 typical antipsychotics and 1 was not prescribed with any antipsychotic in AE group/16 were prescribed atypical antipsychotics, 1 typical antipsychotic and 1 was not prescribed with any antipsychotic in control group ***Previous physical exercise experience:*** N/A Outpatients aged 18–55 years with a diagnosis of schizophrenia spectrum disorder (DSM V).	***Design:*** RCT ENEX (HIIT) ENEX CTRL (Psychoeducation) ***Sample size:*** (n = 51) HIIT: n = 17/ENEX: n = 16/CTRL: n = 18 ***Intervention period:*** 12 weeks	*Supervised:* Yes *Familiarization:* N/A ***ENEX (HIIT):*** *F:* 3 sessions/week *IL*: number of bouts as long as possible). X > 105% FTP + 1–3 min < 91% FTP *Ti*: number of bouts up to participants expended 150 kJ *Ty:* cycling performance *V:* until reaching 150 kJ (not time, sets or reps reported) *P:* N/A ***ENEX***: *F:* 3 sessions/week *IL:* <91% FTP *Ti:* X min until reaching 150 kJ *Ty:* cycling performance *V:* until reaching 150 kJ (not time, sets or reps reported) *P:* N/A ***CTRL (Psychoeducation):*** *F:* 3 sessions/week *Ti:* 15–30 min/session *Ty:* mental and physical health content was delivered to the participants *Attrition of the study:* 15.69% (2 drop out the study before participating in the interventions and 6 during the intervention) *Adherence of the study:* N/A	*Analysis within group* ***ENEX (HIIT):***↑ procedural memory consolidation (sleep-dependent memory consolidation) (*p* < 0.001) ↑ logical memory (24 h delayed recall) (*p* < 0.001) = PANSS total score (*p* = 0.548) = positive symptoms (PANSS) (*p* = 0.824) = negative symptoms (PANSS) (*p* = 0.134) ***ENEX:*** ↑ procedural memory consolidation (sleep-dependent memory consolidation) (*p* < 0.05) = logical memory (24 h delayed recall) (*p* = 0.077) = PANSS total score (*p* = 0.460) = positive symptoms (PANSS) (*p* = 0.594) = negative symptoms (PANSS) (*p* = 0.700) ***CTRL:*** = procedural memory consolidation (sleep-dependent memory consolidation) (*p* = 0.023) = logical memory (24 h delayed recall) (*p* = 0.946) = PANSS total score (*p* = 0.806) = positive symptoms (PANSS) (*p* = 0.829) = negative symptoms (PANSS) (*p* = 0.713) *Analysis between groups* ***HIIT vs. CTRL*** HIIT > CTRL at procedural memory consolidation (sleep-dependent memory consolidation) after 12 weeks (*p* < 0.01) HIIT > CTRL at logical memory (24 h delayed recall) after 12 weeks (*p* < 0.05)
Kern et al. [47]	***Sex:*** 94% male (ENEX)/100% male (CTRL) ***BMI* (kg/m^2^):** Intervention group 30.1/control group 30.0 ***Race/Ethnicity:*** Intervention group (66% Black, 11% White, 11% Asian, 11% Hispanic) and control group (67% Black, 17% White, 0% Asian, 11% Hispanic) ***Other diseases:*** yes, chronic diseases ***Pharmacological treatment:*** Yes, antipsychotics 92% in ENEX and 88% in CTRL. ***Previous physical exercise experience***: No (no participation in an aerobic exercise program in the past 6 months) Veterans aged 40–65 with a psychiatric diagnosis of schizophrenia orschizoaffective disorder (DSM V).	***Design:*** RCT ENEX CTRL ***Sample size:*** (n = 53) ENEX: n = 35/CTRL: n = 18 ***Intervention period:*** 12 weeks	*Supervised:* Yes *Familiarization:* N/A ***ENEX:*** *F:* 3 sessions/week *IL:* 60–70% HR_max_ *Ti:* 20–40 min/session *Ty:* progressive ET *V:* 20 min week 1–2, 30 min week 3–4, 40 min week 5–12 *P:* +10 min every 2 weeks ***CTRL:*** *F:* 3 sessions/week *Ti:* 40 min/session *Ty:* stretching exercise. *Attrition of ENEX:* 22.8% *Attrition of CTRL:* 27.8% *Adherence of ENEX:* 81.4% (29.3 of 36 sessions completed) *Adherence of CTRL:* 77.2% (27.8 of 36 sessions completed)	*Analysis within group* ***ENEX:*** = social functioning (*p* = 0.09) = social cognition (*p* = ns) = non-social cognition (*p* = ns) = BPRS scores for positive and negative symptoms (*p* = ns) ***CTRL:*** Not reported significant effects on the outcomes of interest *Analysis between groups* ***ENEX vs. CTRL*** ↑ ENEX vs. ↓ CTRL of social functioning after 12 weeks (d = 0.35, *p* = 0.06)
Huang et al. [48]	***Sex:*** 45.45% male in ENEX/38.23% male in CTRL ***BMI* (kg/m^2^):** 27.6 ± 4.8 in ENEX/26.1 ± 6.1 in CTRL ***Race/Ethnicity:*** N/A ***Other diseases:*** N/A ***Pharmacological treatment:*** use of antipsychotics in 100% of participants, stable doses for at least 1 month before. ***Previous physical exercise experience:*** N/A Patients 20–60 years of age having a diagnosis of schizophrenia (DSM V) with stable psychotic symptoms.	***Design:*** RCT ENEX (aerobic walking + treatment as usual) CTRL (treatment as usual, original lifestyle and psychotropic treatment) ***Sample size:*** (n = 67) ENEX: n = 33/CTRL: n = 34 ***Intervention period:*** 12 weeks	*Supervised:* Yes *Familiarization:* N/A ***ENEX:*** *F:* 3.2 ± 0.8 days /week *IL:* target 40–60% HRR *Ti:* 30–50 min/session *Ty:* walking program *V:* 128.7 ± 29.1 min *P:* N/A ***CTRL (treatment as usual):*** *Ty:* original lifestyle and psychotropic treatment. *Attrition of ENEX:* 15.4% *Attrition of CTRL:* 10.5% *Adherence of groups or study:* N/A	*Analysis within group* ***ENEX:*** ↑ performance verbal memory (time effect, *p* = 0.03; ∆ Z score = 0.52 ± 0.89) ***CTRL:*** ↑ performance verbal memory (time effect, *p* = 0.03; ∆ Z score = 0.47 ± 0.78) *Analysis between groups* *ENEX vs. CTRL* No significant time × group interaction effect on BACS scores or any dimension. ***ENEX: high-intensity vs. low-intensity (cutoff > 40% HRR for high-intensity)*** No significant time × group interaction effect on BACS score. ↑ ENEX high-intensity vs. ↓ ENEX low-intensity, significant time × group interaction effect on verbal fluency (*p* = 0.05) after adjusting for duration of illness (MANCOVA). No significant interaction for the rest of dimensions.
Kimhy et al. [49]	***Sex:*** 63% male (ENEX)/65% male (CTRL) ***BMI* (kg/m^2^):** 31.60 (ENEX)/30.75 (CTRL) ***Race/ Ethnicity:*** 43% Hispanic (ENEX)/29% Hispanic (CTRL) ***Other diseases:*** N/A ***Pharmacological treatment:*** 100% were prescribed antipsychotics and 6% were prescribed beta-blockers. ***Previous physical exercise experience:*** N/A Patients 18–55 years, diagnosis of schizophrenia or related disorders (DSM IV), no changes in the treatment in the last 3 months.	***Design:*** RCT ENEX CTRL (treatment as usual) ***Sample size:*** (n = 33) ENEX: n = 16/CTRL: n = 17 ***Intervention period:*** 12 weeks	*Supervised:* Yes *Familiarization:* N/A ***ENEX:*** *F:* 3 sessions/week *IL:* 60–70% HR_max_ *Ti:* 45 min/session *Ty:* progressive ET + standard psychiatric care. *V:* 135 min per week *P:* +5% HRmax first 4 weeks ***CTRL:*** Standard psychiatric, regular meetings with a psychiatrist, psychologists, social workers, and/or psychiatric nurses. *Attrition of ENEX:* 19% *Attrition of CTRL:* 23.5% *Attrition of the study:* 21% (3 dropped out in ENEX and 4 dropped out in CTRL) *Adherence of ENEX:* 79% (28.5 of 36 sessions) *Adherence of CTRL:* N/A *Adherence of study:* N/A	*Analysis within group* ***ENEX:*** Not reported significant effects on the outcomes of interest ***CTRL:*** Not reported significant effects on the outcomes of interest *Analysis between groups* ***ENEX vs. CTRL*** ↑ENEX (+23%) vs. ↓CTRL (−4.2%) in social functioning index PSRS (*p* = 0.012) No significant differences in social functioning index by SANS (*p* = 0.58) No significant differences in social functioning index by SLOF (*p* = 0.22)
Marzolini et al. [50]	***Sex:*** 51.14% male (CONCEX)/66.66% male (CTRL) ***BMI* (kg/m^2^):** 27.2 ± 1.2 (CONCEX)/29.3 ± 2.2 (CTRL) ***Race/Ethnicity:*** N/A ***Other diseases:*** At least cardiovascular risk. ***Pharmacological treatment:*** 6 used atypical antipsychotics, 5 used typical antipsychotics and 3 used antianxiety in CONCEX and 5 used atypical antipsychotics, 3 used typical antipsychotics, 2 used antidepressants and 4 used antianxiety in CTRL. ***Previous physical exercise experience:*** N/A Individuals with a diagnosis of schizophrenia/schizoaffective (DSM IV) and at least 1 cardiovascular risk.	***Design:*** RCT CONCEX CTRL (usual care) ***Sample size:*** (n = 13) CONCEX: n = 7/CTRL: n = 6 ***Intervention period:*** 12 weeks	*Supervised:* Yes *Familiarization:* N/A ***CONCEX:*** *F:* 2 sessions/week *IL:* RT: starting 60% 1RM. 10–15 reps (last set repetition at RPE 15)/ET: 60–80% HRR (RPE 11–14) *Ti:* 20 min (RT) + 60 min (ET) each session *Ty:* CT *V:* RT: 4 exercises upper- and lower-limbs: 1–2 × 10–15 reps, r: >30 s/ET: 1.6km to 6.4 km *P:* RT: +1–2 kg based on RPE/ET: +3.33% HRR each 2 weeks ***CTRL:*** Usual care *Attrition of study:* 0% (all participants attended, at least, 50% of sessions; no participant dropped-out) *Adherence of CONCEX:* 72% (±4.4%)	*Analysis within group* ***CONCEX:*** ↑ MHI score after 12 weeks (*p* = 0.03) ↑ 6MWD = ↑ MHI total score (*p* = 0.09) ↓ depressive symptoms (MHI subscale) = ↑ 6MWD (*p* < 0.001) ↓ depressive symptoms (MHI subscale) = ↑ adherence to exercise (*p* = 0.02) ***CTRL:*** = MHI score after 12 weeks (*p* = 0.57) ↑ 6MWD = ↑ MHI total score (*p* = 0.09) ↓ depressive symptoms (MHI subscale) = ↑ 6MWD (*p* < 0.001) *Analysis between groups* ***CONCEX vs. CTRL*** No significant differences in MHI score (*p* = 0.33)
Ryu et al. [51]	***Sex:*** 50% male in ENEX/56.66% male in CTRL ***BMI* (kg/m^2^):** N/A ***Race/Ethnicity:*** N/A ***Other diseases:*** N/A ***Pharmacological treatment:*** 100% were prescribed antipsychotics with a stable dose for at least 4 weeks before intervention. ***Previous physical exercise experience:*** No (exclusion of patients who participated in any exercise program 3 months before the study). Outpatients 18–65 years old, with a diagnosis of schizophrenia or schizoaffective disorder (DSM IV).	***Design:*** Single blind RCT ENEX (outdoor cycling) CTRL (occupational therapy) ***Sample size:*** (n = 60) ENEX: n = 30/CTRL: n = 30 ***Intervention period:*** 16 weeks	*Supervised:* Yes *Familiarization:* N/A ***ENEX:*** *F:* 1 session/week *IL:* 16km/h *Ti:* 40 min of bike training each session *Ty:* outdoor cycling. *V:* 40 min per week *P:* N/A ***CTRL:*** *F:* 1 session/week *Ty:* daily living skills, social skills, or creative activities. *Ti:* 90 min/session *Attrition of ENEX:* 13.3% *Attrition of CTRL:* 20% *Attrition of the study:* 16.7% *Adherence of groups or study:* N/A	*Analysis within group* ***ENEX:*** ↓ psychotic symptoms (BPRS) (*p* = 0.042) ↓ thought disturbance (subscales of BPRS) (*p* = 0.002) ↓ BDI score (*p* < 0.001) ↓ STAI-state score (*p* = 0.001) ↓ STAI-trait score (*p* < 0.001) ↑ GAF score (*p* = 0.001) ↑ WCST CR (*p* < 0.001) ↑ WCST CC (*p* = 0.005) ***CTRL:*** = psychotic symptoms (BPRS) (*p* = 0.136) = BDI score (*p* = 0.945) = STAI-state score (*p* = 0.696) = STAI-trait score (*p* = 0.788) = GAF score (*p* = 0.556) = WCST CR (*p* = 0.406) = WCST CC (*p* = 0.838) *Analysis between groups* ***ENEX vs. CTRL*** No significant group × time interaction in RSES (*p* = 0.052) No significant group × time interaction in QoL score (*p* = 0.098)
Battaglia et al. [52]	***Sex:*** 100% male ***BMI* (kg/m^2^):** 28.55 ± 4.06 in ENEX/28.65 ± 2.62 in CTRL ***Race/ Ethnicity:*** N/A ***Other diseases:*** N/A ***Pharmacological treatment:*** 100% were prescribed antipsychotics (clozapine, olanzapine or risperidone) with a stable dose. ***Previous physical exercise experience:*** at least 1 year of soccer experience. Male patients with a diagnosis of schizophrenia or schizoaffective disorders (DSM IV) > 18 years old.	***Design:*** Double- blind RCT ENEX CTRL ***Sample size:*** (n = 18) ENEX: n = 10/CTRL: n = 8 ***Intervention period:*** 12 weeks	*Supervised:* Yes *Familiarization:* N/A ***ENEX:*** *F:* 2 sessions/week *IL:* 50–85% HR_max_ *Ti:* 40–60 min of training period each session *Ty:* progressive soccer technical-tactical exercises and soccer games *V:* 80–120 min per week *P:* +5 min each game after week 5 and +5 min each game after week 8. ***CTRL:*** not performed any organized physical activity. *Attrition/Adherence:* N/A	*Analysis within group* ***ENEX:*** ↑ MCS-12 score (+10.8%, *p* < 0.0001) ***CTRL:*** Not reported significant effects on the outcomes of interest *Analysis between groups* ***ENEX vs. CTRL*** ENEX > CTRL in MCS-12 score after 12 weeks (*p* < 0.0001)
Nygård et al. [53]	***Sex:*** 58.33% male ***BMI* (kg/m^2^):** ~29.7 (CONCEX)/~30 (CTRL) ***Race/Ethnicity:*** N/A ***Other diseases:*** 11 smokers in TG and 8 in CG (without any other chronic disease). ***Pharmacological treatment:*** 24 were prescribed antipsychotics, 3 antiepileptics, 8 benzodiazepine, 2 biperiden, 1 levaxine, 1 lithium and 1 SSRI in ENEX/21 were prescribed antipsychotics, 3 antiepileptics, 6 benzodiazepine, 1 levaxine, 2 lithium and 3 SSRI in CTRL. ***Previous physical exercise experience:*** N/A Outpatients, 22–59 years old with a diagnosis of schizophrenia spectrum disorders (ICD-10).	***Design:*** RCT CONCEX CTRL ***Sample size:*** (n = 36) CONCEX: n = 17/CTRL: n = 19 ***Intervention period:*** 12 weeks	*Supervised:* Yes *Familiarization:* 1 day with a familiarization session on treadmill and leg press. ***CONCEX:*** *F:* 2 sessions/week *IL:* RT: 90% of 1RM/ET: 85–95% of HR_peak_ *Ti:* 35 min (ET) *Ty:* endurance interval training on treadmill + leg press MST *V:* RT: 4 × 4 reps, r: 3–4 min. 8 reps per session/ET: 4 × 4 min, r: 3 min (at 70% of HR_peak_). 70 min per week *P:* +5 kg each session the patient managed to complete 5 reps (RT) ***CTRL:*** 2 training sessions; participants were encouraged to train on their own. *Attrition of ENEX:* 32% *Attrition of CTRL:* 17% *Adherence of groups or study:* N/A	*Analysis within group* ***CONCEX:*** = mental health index by SF-36 (*p* = 0.158) ***CTRL:*** = mental health index by SF-36 (*p* = 0.934) *Analysis between groups* ***CONCEX vs. CTRL*** No group × time significant differences in mental health index by SF-36 (*p* = 0.277)
Massa et al. [54]	***Sex:*** 76.19% male (ENEX) /88.23% male (CTRL) ***BMI* (kg/m^2^):** 31.11 ± 6.87 (ENEX)/29.31 ± 4 (CTRL) ***Race/Ethnicity:*** 19 African American (ENEX)/17 African American (CTRL). ***Other diseases:*** N/A ***Pharmacological treatment:*** 14 were prescribed atypical antipsychotics, 1 typical antipsychotics, 2 both, 6 antidepressants and 4 were not prescribed with any antipsychotic in ENEX/ 13 were prescribed atypical antipsychotics, 2 typical antipsychotics, 1 both, 4 antidepressants and 1 were not prescribed with any antipsychotic in CTRL. ***Previous physical exercise experience:*** No, sedentary lifestyle for the last month. Outpatients with a diagnosis of schizophrenia, 18–70 years old.	***Design:*** RCT (assessments were completed by an assessor-blinded to the treatment group): ENEX CTRL ***Sample size:*** (n = 38) (only completed 15) ENEX: n = 21 (completed the study 9) CTRL: n = 17 (completed the study 6) ***Intervention period:*** 12 weeks (follow-up to week 20)	*Supervised:* Yes *Familiarization:* N/A ***ENEX:*** *F:* 3 sessions/week *IL:* 50–80% of HR_max_ *Ti:* 20–45 min *Ty:* progressive ET on a stationary bicycle ergometer. *V:* from 60 min to 135 min per week *P:* +5 min and +5% HR_max_ per week ***CTRL:*** 3 sessions/week *Ty:* stretching and toning exercise performed for the same amount of time as the AE program. *Attrition of ENEX:* 68.08% *Attrition of CTRL:* N/A *Adherence of groups or study:* N/A	*Analysis within group* ***ENEX:*** No significant difference after week 12 for MCCB composite scores ***CTRL:*** No significant difference after week 12 for MCCB composite scores *Analysis between groups* ***ENEX vs. CTRL*** ↑ENEX vs. ↓CTRL in MCCB composite score from week 12 to week 20 (*p* = 0.03) ↑ENEX vs. ↓CTRL in visual learning domain of MCCB composite score from week 12 to week 20 (*p* = 0.006)
Su et al. [55]	***Sex:*** 45.5% male (ENEX)/45.5% male (CTRL) ***BMI* (kg/m^2^):** 30.72 (ENEX)/34.56 (CTRL) ***Race/ Ethnicity:*** N/A ***Other diseases:*** N/A ***Pharmacological treatment:*** were on stable antipsychotic medication with no major dose changes for at least 3 months before the study. ***Previous physical exercise experience:*** N/A Patients with a diagnosis of criteria for schizophrenia or schizoaffective disorder (DSM IV) with a stable medication.	***Design:*** Single-blinded RCT ENEX CTRL (stretching and toning control group) ***Sample size:*** (n = 44) ENEX: n = 22/CTRL: n = 22 ***Intervention period:*** 12 weeks (follow-up to 6 months)	*Supervised:* Yes *Familiarization:* N/A ***ENEX:*** *F:* 4–5 sessions/week *IL:* 55–69% HRmax/13–16 RPE *Ti:* 30 min *Ty:* progressive ET on treadmill *V:* 120–150 min per week *P:* Not specified. ***CTRL:*** *F:* 4–5 sessions/week *IE:* 14 exercises, held for 10 s and repeated 10 times. *Ti:* 30 min *Ty:* stretching and toning control program. *Attrition of groups or study:* N/A *Adherence of ENEX:* 76.6% (45.95 of the 60 maximum scheduled sessions) *Adherence of CTRL:* 78.2% (46.91 of the 60 maximum scheduled sessions)	*Analysis within group* ***ENEX*** ↑ processing speed scores at posttest (*p* = 0.005) ↑ processing speed scores at follow-up (*p* = 0.009) ↑ attention scores at follow-up > posttest (*p* = 0.006) ↑ verbal learning scores at follow-up > posttest (*p* = 0.009) ↑ verbal learning scores at follow-up > pretest (*p* = 0.3001; ↑25.6%) ***CTRL:*** Processing speed scores at follow-up > posttest (*p* = 0.02) ↑ verbal learning scores at posttest (*p* = 0.02) Reasoning and problem solving at follow-up > posttest (*p* = 0.003) *Analysis between groups* *ENEX vs. CTRL* ENEX > CTRL In processing speed scores at posttest (*p* = 0.001) ENEX > CTRL in attention scores at posttest (*p* = 0.03) CTRL > ENEX in PANSS negative symptoms score at follow-up (*p* = 0.03)
Kimhy et al. [56]	***Sex:*** 63% male (ENEX)/65% male (CTRL) ***BMI* (kg/m^2^):** 31.60 (ENEX)/30.75 (CTRL) ***Race/Ethnicity:*** 43% Hispanic (ENEX)/29% Hispanic (CTRL) ***Other diseases:*** 25% smokers (ENEX)/23% smokers (CTRL) ***Pharmacological treatment:*** 100% used antipsychotics, 44% used antidepressants and 31% used SSRIs in ENEX/100% used antipsychotics, 35% used antidepressants and 23% used SSRIs in CTRL. ***Previous physical exercise experience:*** N/A Outpatients with a diagnosis of schizophrenia or related disorders (DSM IV); age 18–55 years.	***Design:*** Single-blind RCT ENEX CTRL (treatment as usual) ***Sample size:*** (n = 33) ENEX: n = 16/CTRL: n = 17 ***Intervention period:*** 12 weeks	*Supervised:* Yes *Familiarization:* N/A ***ENEX:*** *F:* 3 sessions/week *IL:* 60–75% of HR_max_ *Ti:* 45 min *Ty:* progressive ET with 2 active-play video game systems, 2 treadmill machines, a stationary bike and an elliptical machine. *V:* 135 min per week *P:* +5% of HR_max_ after week 1, +5% after week 2 and +5% after week 3. ***CTRL:*** Standard psychiatric care, regular meetings with a psychiatrist, psychologists, social workers or psychiatric nurses. *Attrition of ENEX:* 18.7% (dropped-out 3) *Attrition of CTRL:* 23.5% (dropped-out 4) *Attrition of the study:* 21% *Adherence of ENEX:* 79% (28.5 of 36 sessions) *Adherence of study:* N/A	*Analysis within group* ***ENEX:*** Not reported ***CTRL:*** Not reported *Analysis between groups* *ENEX vs. CTRL* ↑ENEX (+15%) > ↓CTRL (−2%) in MCCB composite scores after 12 weeks (*p* = 0.031)
Nygård et al., 2023 [57]	***Sex***: 58.33% male ***BMI (kg/m^2^):*** 27.98 ± 1.67 (CONCEX), 25.5 ± 5.0 (CTRL) ***Race/Ethnicity:*** N/A ***Other diseases:*** N/A ***Pharmacological treatment:*** First generation antipsychotics (n = 7), second generation antipsychotics (n = 42), clozapine (n = 18), 2 antipsychotics (n = 17) and without antipsychotics (n = 3) ***Previous physical exercise experience:*** N/A Outpatients with a schizophrenia spectrum disorders diagnosis (International Statistical Classification of Diseases (ICD)-10) between 18 and 65 years with clinically stable disease.	***Design***: RCT CONCEX CTRL ***Sample size:*** (n = 48) CONCEX: n = 25/CTRL: n = 23 ***Intervention period:*** 1 year	*Supervised:* Yes *Familiarization:* 1 session ***CONCEX***: 2 sessions/week *IL*: RT: 90% 1RM/ET: 85–95% HR_peak_ *Ti*: RT: N/A/ET: 35 min *Ty*: leg press MST + interval ET. *V*: RT: 4 × 4 reps of leg press (r: 3–4 min), 32 reps per week/ET: 4 × 4 min treadmill walking/running (r: 3 min) *P*: RT: +5 kg if 5 reps in last set/ET: +1.75% per week *CTRL*: 2 introductory training sessions to inform them of the benefits of regular exercise and encourage them to train on their own. *Attrition of CONCEX:* 40% *Attrition of CTRL:* 21.74% *Adherence of study:* 64.58% (62 ± 16 attended sessions of 96)	*Analysis within group* ***CONCEX:*** = SF-36 score after 3 months (*p* = 0.720) = SF-36 score after 1 year (*p* = 0.336) ***CTRL:*** = SF-36 score after 3 months (*p* = 0.720) = SF-36 score after 1 year (*p* = 0.336) *Analysis between groups* ***CONCEX vs. CTRL*** No group × time significant differences in SF-36 scores after 3 months (*p* = 0.436) or 1 year (*p* = 0.304)

RCT: randomized controlled trial; ENEX: endurance exercise; RESEX: resistance exercise; CONCEX: concurrent exercise; CTRL: control group; ET: endurance training; RT: resistance training; CT: concurrent training; HIIT: high-intensity interval training; FTP: functional threshold power; MST: maximal strength training; IL: intensity load; Ti: time; Ty: type; V: volume; P: progression; ns: not significant; DSM: diagnostic and statistical manual of mental disorders; PANSS: positive and negative syndrome scale; SF-36: short form-36 health survey; BPRS: brief psychiatric rating scale; BACS: brief assessment of cognition in schizophrenia; PSRS: positive self-relation scale; SANS: scale for the assessment of negative symptoms; SLOF: specific level of functioning; 6MWD: six minute walk distance; MHI: mental health inventory; BDI: Beck’s depression inventory; RSES: Rosenberg self-esteem scale; STAI: state-trait anxiety inventory; GAF: global assessment of functioning; QoL: quality of life; WCST-CC: Wisconsin card sorting test-categories complete; WCST-WR: Wisconsin card sorting test-correct rate; MCS-12: mental component summary; SF-12: short form-12 health survey; MCCB: MATRIC’s consensus cognitive battery; N/A: not applicable.

## 3. Results

### 3.1. Study Selection

The electronic search strategy retrieved 1210 articles, of which 480 papers stemmed from Web of Science, 442 from Scopus, 266 from PubMed, and 22 from SPORTDiscus with full text. After removing duplicates, 696 potentially relevant studies remained for screening. After analyzing the title and abstract, 655 studies were excluded and 41 full-text articles were reviewed (Appendix A). Finally, 13 of those papers were relevant to this review. The PRISMA flow diagram is shown in Figure 1.

### 3.2. Study Characteristics

The main characteristics and findings of interest are shown in Table 2. The twelve eligible articles included a total sample of 553 participants over 18 years old [45,46,47,48,49,50,51,52,53,54,55,56,57]. All of these studies were randomized controlled trials, of which only seven were blinded [49,51,52,54,55,56,57]. According to the type of training, most articles evaluated interventions with endurance training (ET) (70%) [46,47,48,49,51,52,54,55,56]. The other studies used concurrent or combined training (CT) programs as exercise intervention [45,50,53,57], and one of them additionally implemented an isolated resistance exercise training (RT) intervention [45]. Most of these protocols lasted 12 weeks [46,47,48,49,50,52,53,54,55,56] with the exception of three studies that used 20 weeks [45], 16 weeks [51], and 1 year [57], respectively.

### 3.3. Critical Appraisal

#### 3.3.1. Risk of Bias

The risk of bias reported by the Rob 2 tool is shown in Figure 2 and Figure 3. Nine (70%) of the included articles were considered at *high risk* of bias in the overall score [45,46,47,49,51,52,53,54,55]. Concerning the different domains, eleven studies (84.6%) were assessed as being at *low risk* regarding bias arising from the randomization process [45,46,48,50,51,52,53,54,55,56,57]. Only four randomized controlled trials (30.77%) were judged as being at *low risk* considering bias due to deviations from intended interventions [48,49,56,57]. Seven studies (53.84%) reported a *low risk* of bias due to missing outcome data [46,47,48,49,50,56,57]. All studies except one [53] were classified as *low risk* in the fourth domain (bias in the outcome measurement). Finally, five studies (30.77%) were assessed as being at *low risk* regarding bias in the selection of the reported results [45,48,53,56,57].

#### 3.3.2. Quality Reporting of Exercise Intervention in the Training Programs

The rigor of the intervention protocols regarding the total sample (TS), ET, CT, and RT is shown in Table 3 after the evaluation of the CERT items in the included articles. The reviewers evaluated item 13, which refers to the description of the exercise, with a specific emphasis on whether the exercise fulfilled the F.I.T.T.−V.P. parameters. However, the articles did not provide a complete account of all the parameters of F.I.T.T.−V.P. The aforementioned item 13 was the most commonly reported in exercise interventions (100%). The second most frequently reported item was whether the exercises were generic or tailored to the individual, with 76.9%, 66.7%, 100%, and 100% reporting for TS, ET, CT, and RT, respectively. The third most commonly reported item was the degree to which the intervention was executed according to plan, with 76.9%, 77.7%, and 66.7% reporting for TS, ET, and CT, respectively; however, this item was not reported in the resistance training protocol. On the other hand, *adherence*, *replication,* and *home components* were not reported by any of the included articles, while *motivation strategies* were reported in less than 10% of the selected studies. Further, only Kimhy et al. [49] reported more than 50% of CERT items when describing their exercise intervention.

### 3.4. Summary of Evidence

#### 3.4.1. Psychopathology

Eight articles among the eligible studies evaluated the effects of physical exercise on the psychopathology of individuals diagnosed with schizophrenia [45,46,47,49,50,51,52,55]. The measuring instruments used to evaluate the results were the positive and negative symptoms scale (PANSS) [45,46,49,55], the mental health inventory (MHI) [50], the brief psychiatric rating scale (BPRS) [47,51], Beck’s depression inventory (BDI) [51], the state and trait anxiety inventory (STAI) [51], the Rosenberg self-esteem scale (RSES) [51], and the mental component summary (MCS-12) of self-reported QoL short form (SF-12) [52]. 

Taking all these results together, most of the reviewed studies showed benefits in some psychopathological variables after exercise independently of the training protocol [45,50,51,52,55]. Only one of the studies used two training protocols with different types of exercise (i.e., ET and RT), demonstrating a decrease in negative symptoms only after RT [45]. Three of the included studies [46,47,49] found no significant differences in positive or negative symptoms of schizophrenia following an ET program with comparable characteristics (i.e., three sessions per week over 12 weeks, with a duration of 20 to 60 min per session). 

#### 3.4.2. Cognition

Eight of the eligible articles assessed cognitive variables using different measuring instruments. In this regard, Kimhy et al. [56], Su et al. [55], and Massa et al. [54] revealed improvements in MATRICS Consensus Cognitive Battery (MCCB) scores in patients with diagnosis of schizophrenia. Specifically, these studies found improvements in processing speed, attention, verbal learning domains [55], and visual learning [54] following a regular ET program. Huang et al. [48] found no significant improvements in cognitive function as assessed using the Brief Assessment of Cognition in Schizophrenia (BACS) instrument when comparing the exercise group and control group (maintenance of original lifestyle and psychotropic treatment) following a 12 week walking exercise program (i.e., ET). Ryu et al. [51] reported an increase in executive function scores as measured by the Wisconsin Card Sorting Test (WCST) in individuals diagnosed with schizophrenia after following an outdoor cycling training. In the same line, Lo et al. [46] evaluated sleep-dependent procedural memory consolidation performance as measured by the finger-tapping motor sequence task (MST). These authors showed that both high-intensity interval training (HIIT) and moderate-to-vigorous (91–105% of the functional threshold power, FTP) ET had a positive impact on memory consolidation. Moreover, HIIT was found to be more effective than moderate-to-vigorous (91–105% FTP) continuous ET.

#### 3.4.3. Quality of Life

A number of recent studies [45,51,52,53,57] have used different instruments to assess patients’ QoL (e.g., the Short Form (SF)-36 Health Survey, World Health Organization Quality of Life Scale Abbreviated Version (WHOQOL-BREF), SF-12 Health Survey, Patient Activation Measure-13). However, the impact of exercise programs on the QoL of individuals with schizophrenia remains inconclusive. Silva et al. [45] and Battaglia et al. [52] reported positive effects of physical exercise, whereas other studies [51,53,57] did not obtain significant changes. Conversely, with regard to the social domain, Kern et al. [47] and Kimhy et al. [49] demonstrated that physical exercise could benefit social functioning.

## 4. Discussion

The main findings of this scoping review were: (1) supervised regular physical exercise of various modalities improves the psychopathology of schizophrenia; (2) regular ET seems to be an effective strategy to enhance cognitive function in patients with schizophrenia, while evidence regarding other training modalities is limited; and (3) while regular exercise may promote both social functioning and social cognition, thereby positively affecting the QoL of patients diagnosed with schizophrenia, the results are currently inconclusive.

To the best of our knowledge, this is the first scoping review to include studies that meet strict selection criteria for ensuring replicability regarding exercise training protocol characteristics. Further, a special focus was placed on the effects of those training protocols on the three main non-physical health-related areas affected by schizophrenia (i.e., psychopathology, cognition, and QoL). Considering the limited availability of randomized controlled trials satisfying the predetermined selection criteria, conducting a meta-analysis of the data was unfeasible.

### 4.1. Training Characteristics of the Included Studies

Concerning the training protocol duration, most of the works reported an intervention period of 12 weeks [46,47,48,49,50,52,53,54,55,56]. However, three studies deviated from this standard duration, with intervention periods of 16 weeks [51], 20 weeks [45], and 1 year [57]. Furthermore, other reviews found a similar mean duration [58,59], while certain authors [31,32] presented heterogeneous durations in their included studies. In this context, extended intervention durations exhibited a correlation with reduced rates of adherence to training programs. This implies that briefer intervention periods may be better tolerated by individuals diagnosed with schizophrenia. Additional research is needed to explore strategies aimed at improving adherence to exercise interventions in individuals with schizophrenia. It is worth noting that, concerning the form of physical exercise utilized, the prevailing mode of physical training in this review was ET [46,47,48,49,51,52,54,55,56]. Only one of the selected studies included RT for one of its groups of participants [45]. Our results are in accordance with most previous reviews, as the majority of their articles reported ET interventions [32,58,59,60]. In line with this, in a recent review by Bredin et al. [33], the above authors refer to three studies reporting RT interventions. However, when analyzing this more deeply, one of the studies is already included in our review, while the others are a pilot study [61] and a study with an intervention protocol that was not described in sufficient detail to ensure replication [62]. Although scientific evidence has widely demonstrated that RT has important health-related benefits on the regulation of diabetes, weight loss, and improvements in cardiovascular health [63,64], as well as in cognitive abilities and self-esteem [63], RT interventions among patients with schizophrenia are scarce. 

Regarding the quality of the analyzed training protocols, it was observed that many of the CERT tool items were not reported. It is striking that less than 10% of the studies reported the motivational strategies implemented in their training interventions. In this regard, it has been previously suggested that patients suffering from schizophrenia present different barriers that prevent them from reaching the minimum exercise recommendations, with lack of motivation being one of the most important [65,66,67]. Hence, it is of paramount importance to describe in detail those motivational strategies that optimize adherence to physical exercise in patients with schizophrenia [66,67]. While recent systematic reviews have neglected to incorporate the CERT checklist for the reviewed articles, and have further failed to report data pertaining to motivation strategies and adherence to the exercise sessions [36,37,39,40], Gallardo et al. [37] and Cella et al. [36] have underscored the pivotal role of motivation in ameliorating adherence to interventions and optimizing the positive effects of exercise programs among this population. 

In the reviewed body of literature, it is noteworthy that only approximately 50% of the studies reported on the inclusion of non-exercise components. The non-exercise component encompasses social interactions and interpersonal relationships, both of which constitute integral elements of supervised training and group-based interventions. It is imperative to acknowledge that these facets may exert an influence on the outcomes observed after physical exercise interventions [66].

### 4.2. Effects of Regular Exercise on Psychopathology, Cognition, and Quality of Life of Patients with Schizophrenia

#### 4.2.1. Psychopathology

Concerning the effects of overall physical exercise on patients with schizophrenia, the results of the included studies reported important health-related benefits. Silva et al. [45], Marzolini et al. [50], and Ryu et al. [51] showed improvements in psychiatric symptoms of the disease. These findings align with previous reviews [30,37,58] that reported positive outcomes of ET and other training modalities on both the positive and negative symptoms of schizophrenia, validating the beneficial effects of these interventions. In this regard, another recent review by Bredin et al. [33] showed improved PANSS scale scores following walking training. Further, the authors of a study that implemented an aerobic exercise program for 12 weeks demonstrated that exercise both improved patients’ symptoms and prevented up to a 7% increase in negative symptoms observed in the subsequent follow-up of patients in the control group [68]. Similar results have been reported by several studies [30,31,35,60,69,70], whereas other authors [46,47,49] did not show significant changes in psychiatric symptoms after prescribing ET. In this regard, a minimum physical exercise intensity threshold seems to be necessary for obtaining health benefits, as when the data related to low-intensity interventions (e.g., yoga and stretching) were removed from the meta-analysis these authors obtained significant improvements in psychiatric symptoms [32]. Additionally, interventions involving moderate-to-vigorous exercise accumulating approximately 90 min per week produced significant reductions in psychiatric symptoms and improved the cognitive function, co-morbid disorders, and functioning of individuals with schizophrenia [32]. It is difficult to analyze what the causes of these discrepancies in these results might be. On the one hand, there is some difficulty in carrying out the methodology of studies with physical exercise in patients with schizophrenia. We can see this reflected when looking at the dropout rates of the included studies. Despite the strictness of the inclusion criteria in terms of methodology, most studies exceeded a 10% dropout rate [45,46,47,48,49,51,52,53,54,55,56]. Specifically, one exceeded a 50% dropout rate without specifying any medical or personal cause that prevented the patients from carrying out exercise [54]. The differences between patients and the wide variety of physical, psychological, and social factors that impact their lives may be among the reasons that hinder the generalization of results [69]. Along with this, the methodological shortcomings of the included studies and their exercise interventions, which we have previously analyzed, make it difficult to compare results [28]. 

Focusing attention on affective symptoms reported in the included articles, Marzolini et al. demonstrated improvements in depressive symptoms by enhancing functional capacities after 12 weeks of CT [50]. Similarly, Ryu et al. [51] showed improved depressive symptoms and levels of anxiety after 16 weeks of outdoor cycling in patients with schizophrenia. This association between aerobic exercise and depressive symptoms in patients with schizophrenia was already demonstrated in 1993 by Pelham et al. [71], who concluded that this type of exercise decreased depressive symptoms. Furthermore, more recent studies support these positive effects after 12 weeks of CT [72], while other authors have reported positive effects on anxiety [73].

#### 4.2.2. Cognition

Regarding the cognitive effects of physical exercise on schizophrenia, our results are in accordance with previous reviews [59,74]. Firth et al. [59] additionally differentiated those domains that independently reflect a significant improvement after the exercise intervention (i.e., working memory, social cognition, and attention/vigilance). In these domains, we found specific improvements in three of the clinical trials selected in our review [47,48,55]. In this regard, a recent systematic review and meta-analysis performed with older adults suggested a nonlinear dose-response relationship between overall exercise and cognition [75]. The authors estimated a minimal threshold of 724 METs a minute per week for obtaining relevant changes in cognition. Moreover, although lower doses of different exercise types could already produce benefits, it seems that RT leads to greater positive effects on cognition in older adults than other training modalities [75]. Certain authors have mentioned the relationship between training intensity and the magnitude of effects on cognition in patients with schizophrenia [48]. These researchers found study greater benefits in attention, processing speed, and verbal fluency with high ET intensity [48]. Nevertheless, further investigation is warranted in order to assess the potential of resistance training (RT) as a cognitive enhancement intervention in individuals diagnosed with schizophrenia.

#### 4.2.3. Quality of Life and Functioning

The majority of research included in the present scoping review supports the implementation of physical exercise as a tool to improve the QoL and functioning of individuals with schizophrenia [45,49,52]. Nevertheless, further studies are warranted to meticulously address potential biases and provide comprehensive descriptions of the independent variable, that is, exercise, in order to enhance the quality of evidence in this field. In this regard, three studies that investigated QoL and functioning did not find significant improvements in the different scales used [47,53,57]. 

Ryu et al. [51] showed improvements in global functioning index as measured by the GAF scale after 16 weeks of ET; however, they did not report changes in QoL. These results are in line with a pilot study by Loh and colleagues [70], who showed improvements in QoL, the social functioning component, and the global functioning index as assessed through the personal social performance scale (PSP) after 12 weeks of structured walking training. According to this, Acil et al. [24] observed improvements after an ET program in the scores related to QoL in participants with schizophrenia. Although there is some controversy surrounding this topic, and the results obtained from various studies remain inconclusive and subject to methodological limitations, the available evidence suggests that regular exercise has the potential to enhance both social and global functioning, exerting a positive impact on the QoL of individuals diagnosed with schizophrenia.

Considering all the evidence, the regular engagement in diverse forms of physical exercise exhibited potential in yielding health benefits in the mental and social domains in patients diagnosed with schizophrenia. Nevertheless, there is a lack of knowledge on the optimal dose–response relationship and the types of exercise that produces the greatest health benefits and optimizes QoL.

## 5. Future Perspectives

Due to the nature of this scoping review, the possibility of conducting a meta-analysis was impeded by the inclusion of multiple outcomes and the limited number of articles per outcome. One of the challenges encountered during the review process was the insufficient availability of detailed information regarding exercise characteristics in the reviewed studies, which restricts the transferability of the results to clinical practice and the replicability of future studies.

While recent investigations suggest that multimodal or concurrent training (CT) programs are more effective than isolated exercise training for patients diagnosed with schizophrenia [33,34], further research is needed to establish optimal dose–response relationships and exercise types, particularly in relation to non-physical health-related components such as psychopathology, cognition, and quality of life. Future interventions targeting individuals diagnosed with schizophrenia should adhere to the CERT criteria, which consider factors such as motivation, exercise supervision, adherence, adverse event reporting, and non-exercise components. On the other hand, it could be interesting for future research to consider the influence of hormones on this disease and on the effects of its treatment [76]. In light of the limitations imposed by the challenges of daily life associated with this condition, considering these aspects could improve both the methodological approach and the training protocol itself, leading to overall enhancements in patients diagnosed with schizophrenia.

## Figures and Tables

**Figure 1 behavsci-13-00959-f001:**
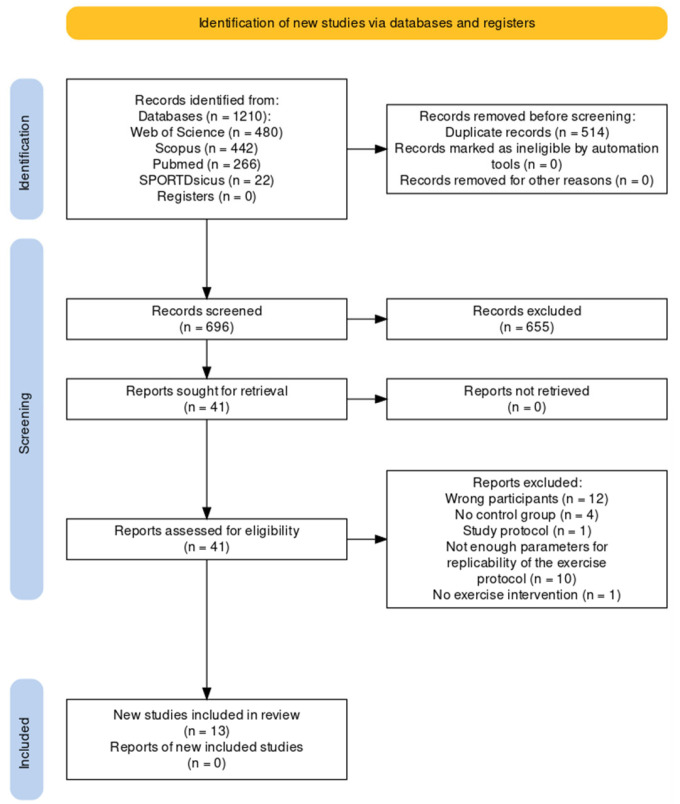
Flowchart of study following the PRISMA (Preferred Reporting Items for Systematic Meta-Analyses) guidelines.

**Figure 2 behavsci-13-00959-f002:**
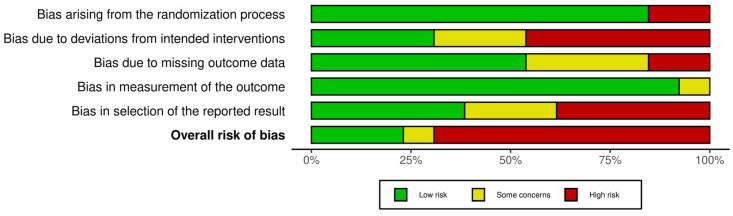
Risk of bias graph: judgment regarding each risk-of-bias item, presented as percentages across all included studies.

**Figure 3 behavsci-13-00959-f003:**
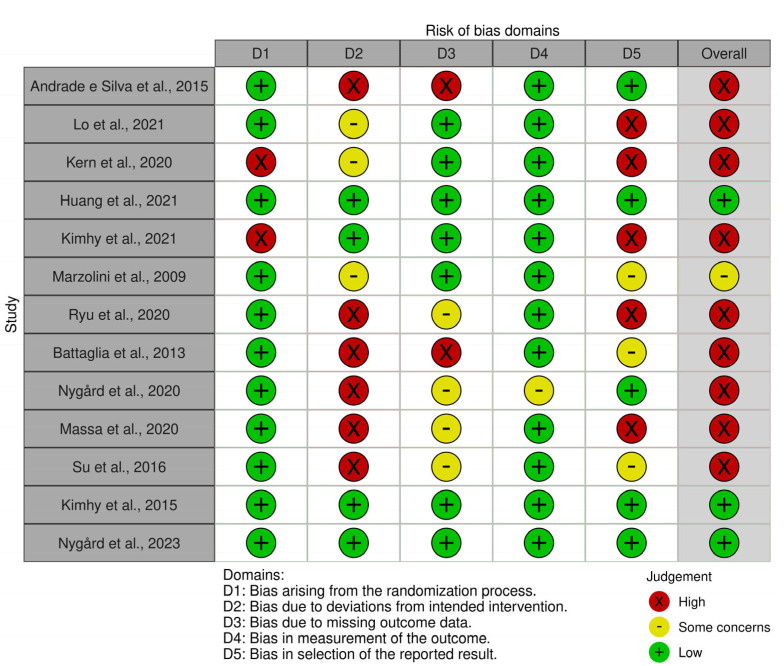
Ratings considering risk-of-bias items for each study. The references of the cited articles [45,46,47,48,49,50,51,52,53,54,55,56,57] can be found in the reference list.

**Table 1 behavsci-13-00959-t001:** Inclusion and exclusion criteria according to PICOS.

PICOS Category	Inclusion Criteria	Exclusion Criteria
Population	Participants >18 years and diagnosed with schizophrenia in a non-residential environment.	While including participants with schizophrenia, did not perform a differentiated sub-group analysis.
Intervention	Studies whose exercise protocol had specified, at least, the first five parameters to configure the exercise dose (i.e., F.I.T.T.−V.P.): frequency, intensity, type, time (duration) and volume. Furthermore, the training period must last, at least, three weeks.	Exercise interventions whose protocols were unstructured or did not report, at least, the first five parameters of F.I.T.T.−V.P. previously mentioned.
Comparator	One control group (i.e., not exposed to a regular physical exercise program).	No control group or a control group with an active intervention.
Outcomes	Data evaluating adaptations of regular exercise interventions on psychopathology, QoL or cognition in patients diagnosed with schizophrenia.	No reported measures of psychopathology, QoL or cognition in patients diagnosed with schizophrenia.
Study design	Experimental studies with randomized participants.	Non-experimental and/or non-randomized studies.

**Table 3 behavsci-13-00959-t003:** Exercise intervention quality report of the included studies according to the CERT.

CERT Item	TS (n = 13) n (%)	ET (n = 9) n (%)	CT (n = 4) n (%)	RT ^†^ (n = 1) n (%)
Item 1. What (materials)	9 (69.2%)	6 (66.66%)	3 (75%)	1 (100%)
Item 2. Who (provider)	3 (23%)	3 (33.33%)	0 (0%)	0 (0%)
Item 3. Individually or in a group	6 (46.2%)	5 (55.55%)	1 (25%)	0 (0%)
Item 4. Supervised or unsupervised	6 (46.2%)	5 (55.55%)	1 (25%)	0 (0%)
Item 5. Adherence report	0 (0%)	0 (0%)	0 (0%)	0 (0%)
Item 6. Motivation strategies	1 (7.7%)	1 (11.11%)	0 (0%)	0 (0%)
Item 7a. Exercise progression	6 (46.2%)	3 (33.33%)	3 (75%)	0 (0%)
Item 7b. Program progression	7 (53.8%)	4 (44.44%)	3 (75%)	1 (100%)
Item 8. Exercise replication	0 (0%)	0 (0%)	0 (0%)	0 (0%)
Item 9. Home components	0 (0%)	0 (0%)	0 (0%)	0 (0%)
Item 10. Non-exercise components	4 (30.8%)	4 (44.44%)	0 (0%)	0 (0%)
Item 11. Adverse events report	2 (15.4%)	1 (11.11%)	1 (25%)	0 (0%)
Item 12. Setting	4 (30.8%)	3 (33.33%)	1 (25%)	0 (0%)
Item 13. Description of the exercise	13 (100%)	9 (100%)	4 (100%)	1 (100%)
Item 14a. Exercises generic or tailored?	10 (76.9%)	6 (66.66%)	4 (100%)	1 (100%)
Item 14b. Description of the adaptation made in the exercises	9 (69.2%)	6 (66.66%)	3 (75%)	1 (100%)
Item 15. Rules for starting level	2 (15.4%)	2 (22.22%)	0 (0%)	0 (0%)
Item 16a. How adherence to exercise was measured	5 (38.5%)	5 (55.55%)	0 (0%)	0 (0%)
Item 16b. Is the Intervention carried out according to how it was planned?	10 (76.9%)	7 (77.77%)	3 (75%)	0 (0%)

*TS*: total score; *ET*: endurance training; *CT*: concurrent training; *RT*: resistance training. **^†^** RT’s intervention belongs to an article that also implemented CT [45].

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
