# Peer review of "The Impact of Regular Physical Exercise on Psychopathology, Cognition, and Quality of Life in Patients Diagnosed with Schizophrenia: A Scoping Review"

_behavsci, 2023, doi:10.3390/bs13120959_

Round 1

Reviewer 1 Report

Comments and Suggestions for Authors

This is an interesting and scholarly review. The authors have clearly spent time on this. They have thoroughly investigated the impact of regular physical exercise on psychopathology, cognition, and quality of life in patients with schizophrenia. However, there are several issues to be addressed.

1) You state that this is a scoping review; however, the implementation of strict criteria, the focused analysis of 12 studies and the inclusion of the risk of bias assessment align more closely with the methodologies of a systematic review. It would be beneficial to either adjust the scope and methodology to accurately reflect a scoping review's exploratory nature or reframe the manuscript to reflect a systematic review.

2) Table 1 has a lot of information in it. The information should be simplified so that the table gives a better summary of findings when looking at it at a glance. I would suggest having multiple columns for each of the main participant characteristics. I would also recommend aligning the types of exercise in the last two columns, again to make it easier to read the information (i.e. RESEX in the protocol column starting on the same line as RESEX in the outcomes of interest column). Some of the characteristics, exercise protocol and outcomes of interest could be taken out of the table. Instead, you can describe each study briefly in section 3.2., especially considering there were just 12 eligible articles in your review.

Minor comments:

3) Line 17 should say revealed not reveled.

4) For your keywords used during your search, wouldn’t schizo* already cover schizophrenia, schizoaffective and schizophrenic?

5) I would recommend doing the search again, considering the last search was in 2022.

6) Section 4.2 could be broken down into more subheadings to make it easier for readers.

Comments on the Quality of English Language

Overall, this is well written, though there are some minor grammatical errors throughout the manuscript. Please proofread and fix these.

Author Response

Please see the pdf attached.

King regards.

Reviewer 2 Report

Comments and Suggestions for Authors

Available evidences presented in this review suggest that regular exercise has the potential to enhance both social and global functioning of individuals diagnosed with schizophrenia

I suggest that in the future it would be important to highlight the influence of hormonal regulation (phases of the menstrual cycle) on the pathogenesis and treatment outcomes of schizophrenia

Possible reason of some results obtained from various studies inconsistency behind the inconclusive and subject to methodological limitations, could be unobserved in this review data sex and hormone influences on the schizophrenia severity:

(Papadea D, Dalla C, Tata DA. Exploring a Possible Interplay between Schizophrenia, Oxytocin, and Estrogens: A Narrative Review. Brain Sci. 2023 Mar 8;13(3):461. doi: 10.3390/brainsci13030461)

(da Silva FER, Cordeiro RC, de Carvalho Lima CN, Cardozo PL, Vasconcelos GS, Monte AS, Sanders LLO, Vasconcelos SMM, de Lucena DF, Cruz BF, Nicolato R, Seeman MV, Ribeiro FM, Macedo DS. Sex and the Estrous-Cycle Phase Influence the Expression of G Protein-Coupled Estrogen Receptor 1 (GPER) in Schizophrenia: Translational Evidence for a New Target. Mol Neurobiol. 2023 Jul;60(7):3650-3663. doi: 10.1007/s12035-023-03295-x)

Author Response

Please see the pdf attached.

Kind regards.
